# Isookanin Inhibits PGE_2_-Mediated Angiogenesis by Inducing Cell Arrest through Inhibiting the Phosphorylation of ERK1/2 and CREB in HMEC-1 Cells

**DOI:** 10.3390/ijms22126466

**Published:** 2021-06-16

**Authors:** Yingji Xin, Kyungbaeg Roh, Eunae Cho, Deokhoon Park, Wankyunn Whang, Eunsun Jung

**Affiliations:** 1Biospectrum Life Science Institute, Yongin 16827, Korea; bioua@biospectrum.com (Y.X.); biosh@biospectrum.com (K.R.); biozr@biospectrum.com (E.C.); pdh@biospectrum.com (D.P.); 2Department of Global Innovative Drug, Graduate School, College of Pharmacy, Chung-Ang University, Heukseok-dong, Dongjak-gu, Seoul 156756, Korea

**Keywords:** isookanin, anti-angiogenesis, HMEC-1 cell, prostaglandin E_2_ (PGE_2_), cell cycle arrest, ERK1/2 and CREB signaling pathway

## Abstract

Inflammation is increasingly recognized as a critical mediator of angiogenesis, and unregulated angiogenic responses often involve human diseases. The importance of regulating angiogenesis in inflammatory diseases has been demonstrated through some successful cases of anti-angiogenesis therapies in related diseases, including arthritis, but it has been reported that some synthetic types of antiangiogenic drugs have potential side effects. In recent years, the importance of finding alternative strategies for regulating angiogenesis has begun to attract the attention of researchers. Therefore, identification of natural ingredients used to prevent or treat angiogenesis-related diseases will play a greater role. Isookanin is a phenolic flavonoid presented in *Bidens* extract, and it has been reported that isookanin possesses some biological properties, including antioxidative and anti-inflammatory effects, anti-diabetic properties, and an ability to inhibit α-amylase. However, its antiangiogenic effects and mechanism thereof have not been studied yet. In this study, our results indicate that isookanin has an effective inhibitory effect on the angiogenic properties of microvascular endothelial cells. Isookanin shows inhibitory effects in multiple stages of PGE_2_-induced angiogenesis, including the growth, proliferation, migration, and tube formation of microvascular endothelial cells. In addition, isookanin induces cell cycle arrest in S phase, which is also the reason for subsequent inhibition of cell proliferation. The mechanism of inhibiting angiogenesis by isookanin is related to the inhibition of PGE_2_-mediated ERK1/2 and CREB phosphorylation. These findings make isookanin a potential candidate for the treatment of angiogenesis-related diseases.

## 1. Introduction

Angiogenesis is the formation of new capillaries from existing vasculature and is an integral biological process that occurs throughout life in a number of physiological and pathological processes [1]. It is known that angiogenesis is essential for reproduction, development, and wound repair, where it is highly regulated. However, many diseases are caused by unregulated angiogenesis, including cancer, rheumatoid arthritis, inflammatory bowel disease (IBD), and diabetic retinopathy [2,3]. In addition, angiogenesis is also involved in various skin diseases including psoriasis, telangiectasias, and rosacea, where various types of blood vessel growth can be observed in diseased skin [4].

Many potent proangiogenic factors such as vascular endothelial growth factor (VEGF), basic fibroblast growth factor (bFGF), angiogenin, and transforming growth factor-β (TGF-β) have been found to be involved in angiogenesis [2]. Abnormal angiogenesis caused by these stimulants is accompanied by a series of endothelial cell behavioral changes associated with increased cell proliferation, cell migration and invasion, and the formation of new tubular structures [5]. Angiogenesis inhibitors targeting this set of mechanisms have been studied and applied in clinical trials, where the data have strongly suggested that suppressing aberrant angiogenesis may be a promising treatment for angiogenesis-related diseases [6,7].

Meanwhile, recent clinical and basic scientific studies indicate that angiogenic inducers involve (many) other factors, including inflammatory mediators [8]. Among the many inflammatory mediators, prostaglandins (PGs) are representative and are naturally occurring lipid compounds derived from the cyclooxygenase (COX)-mediated metabolism of arachidonic acid [9]. Prostaglandins link the interactions between various immune regulatory cells and are thought to play a key role in regulating the inflammatory response [10]. Prostaglandin E_2_ (PGE_2_) is the most abundant prostanoid in humans and the most important COX-2 mediated product found to date. PGE_2_ is involved in regulating many different fundamental biological functions, and its role in inflammation and immune response has been well established in previous studies [11,12].

To date, PGE_2_ has been the most researched inflammatory mediator in immune cells, and in recent years, a lot of attention has been paid to the other roles of PGE_2_ [13]. However, the mechanisms and effects of PGE_2_ on endothelial cells and angiogenesis have been less studied [14]. PGE_2_ affects target cells by binding four cognate receptors called EP1, EP2, EP3, and EP4, which belong to a large family of G protein-coupled receptors (GPCRs). Ligand binding of different EP receptors leads to different actions through activation of separate downstream signaling pathways [15]. Some prior studies have confirmed that PGE_2_ and EP receptors have a direct role in angiogenic gene expression and angiogenic cell behavior [14,16,17,18,19].

The importance of angiogenesis in inflammatory disease processes has been demonstrated by successful cases of antiangiogenic therapeutic trials for diseases such as asthma [20], rheumatic diseases [21], and dermatosis [22]. These treatments are commonly used FDA-approved drugs that target angiogenic factors or angiogenic factor signaling cascades [6]; however, these drugs are known to have some clinical limitations and some side effects associated with their administration [23,24].

Therefore, the importance of finding alternative strategies for controlling angiogenesis has begun to attract people’s attention, and the identification of natural compounds will play an important role in finding therapeutics for the prevention and treatment of angiogenesis-related diseases [25]. Isookanin (C_15_H_12_O_6_, molecular weight: 288.3) is a phenolic flavonoid presented in *Bidens* extract [26]. It has been reported that isookanin possesses some biological properties, including antioxidative [27,28] and anti-diabetic properties [27], anti-inflammatory effects [29] and an ability to inhibit α-amylase [26]. However, the antiangiogenic effects and mechanism thereof have not been studied yet.

In this study, we investigated the effect of isookanin on PGE_2_-induced angiogenesis in human dermal microvascular endothelial cells (HMEC-1). Our results show that PGE_2_ led to proliferation of HMEC-1 cells and that isookanin administration suppressed the proliferation, migration, and tube formation ability of HMEC-1 cells. Additionally, the results from the mechanism study showed that isookanin exerted its inhibitory effect on angiogenesis through the induction of cell cycle arrest and the regulation of the PGE_2_ receptor and its downstream ERK1/2 and CREB phosphorylation.

## 2. Results

### 2.1. Effects of Isookanin on PGE_2_-Induced Endothelial Cell Proliferation and Cytotoxicity

Activation of endothelial cell proliferation is one common feature of angiogenesis [30]. To explore the inhibitory effect of isookanin on PGE_2_-induced endothelial cell proliferation in vitro, we performed an MTT assay. HMEC-1 cells were pretreated with isookanin (1, 5, 10 µg/mL) for 2 h and then stimulated with 20 µM PGE_2_ for 48 h prior to evaluation for cell viability. VEGF (100 ng/mL) was used as an angiogenic positive control. Treatment with PGE_2_ alone increased the levels of HMEC-1 cell proliferation, whereas the addition of isookanin inhibited HMEC-1 cell proliferation in a dose-dependent manner (Figure 1A). We also observed via microscopy that treatment with PGE_2_ alone increased the density of endothelial cells, while the addition of isookanin decreased the density of endothelial cells after 48 h, and no cell damage was observed in cell morphology (Figure 1B).

Next, we examined the effect of isookanin on cytotoxicity in HMEC-1 cells to determine whether the antiangiogenic effect was caused by toxicity. Cell cytotoxicity was assessed using an LDH assay, and a significant increase in LDH activity was observed in cells treated with lysis buffer as a high control, showing that this assay system was reliable. In contrast, isookanin showed no increase in LDH activity at concentrations ranging from 1 to 10 μg/mL (Figure 1C) compared with the control. These results indicate that the antiangiogenic activity of isookanin was not caused by cytotoxic actions.

### 2.2. Effects of Isookanin on PGE_2_-Induced Endothelial Cell Migration

Endothelial cell migration is a key process during the formation of new capillaries [31]. Thus, we next investigated the effect of isookanin on PGE_2_-induced endothelial cell migration using a scratch migration assay and a transwell migration assay. The results revealed that the endothelial cells treated only with PGE_2_ showed increased migration, while the cells pretreated with isookanin before PGE_2_ stimulation showed a significant and dose-dependent decrease in cell migration (Figure 2A,B). In the scratch migration assay, endothelial cells stimulated with PGE_2_ increased the cell migration area compared with the non-stimulated group, while treatment with isookanin significantly reduced the cell migration area compared with the group stimulated only with PGE_2_ (Figure 2A). In the transwell migration assay, as shown in Figure 2B, isookanin obviously suppressed endothelial cell migration from the upper chamber to the lower chamber. The inhibitory effect reached 46.9% with 10 µg/mL isookanin compared with the group treated only with PGE_2_ (Figure 2C).

### 2.3. Effect of Isookanin on PGE_2_-Induced Endothelial Cell Tube Formation

In order to further evaluate the antiangiogenic activities of isookanin, we detected the effect of isookanin on the tube formation ability of endothelial cells via tube formation assay, which mimics the angiogenic process. Following seeding onto Matrigel, endothelial cells quickly attached to the matrix and morphologically differentiated to a capillary-like network [32]. As shown in Figure 3A, PGE_2_ could promote the formation of capillary/tube-like networks of HMEC-1 cells, similar VEGF treatment. On the other hand, isookanin-treated HMEC-1 cells formed relatively incomplete and narrow tube-like structures, and this effect became more pronounced with increasing concentration of treatment. In addition, the antiangiogenic activities of isookanin on tube formation were quantified by analyzing the total branch number, length, and mesh size of the tube-like structures using Image-J software. The analysis showed that isookanin at concentrations ranging from 1 to 10 µg/mL significantly reduced the extent of tubular formation of HMEC-1 in a dose-dependent manner (Figure 3A,B).

### 2.4. Effect of Isookanin on HMEC-1 Proliferation Inhibition

To evaluate the anti-proliferative effect of isookanin on PGE_2_-induced HMEC-1 cells, Click-iT EdU incorporation assays were performed, and EdU and DNA contents were analyzed by flow cytometry. Bivariate distributions of EdU incorporation (*y* axis) vs. DNA content (*x* axis) were plotted. The boxes indicate EdU-incorporated cells in the S-phase, and cell proliferations are expressed as percentages of EdU-labeled S-phase populations to total cell numbers. Flow cytometric analysis showed EdU-labeled S phase HMEC-1 cells increased to 19% by PGE_2_ treatment compared with the control. However, the proportion of EdU-labeled cells decreased in a dose-dependent manner after pre-treatment with isookanin and significantly decreased by 85.5% at a concentration of 10 µg/mL (Figure 4A,B). These results once again indicate that isookanin inhibited the proliferation of HMEC-1 cells.

### 2.5. Effects of Isookanin on the Regulation of Cell Cycle and the Expression Level of Cell Cycle Regulatory Proteins

To clarify the relevant mechanism of isookanin inhibiting cell proliferation, we examined the effect of isookanin on the HMEC-1 cell cycle by propidium iodide staining (Figure 5A). The results show that in the group treated only with PGE_2_, there were 46.2% cells in G0/G1 phase, 21.4% cells in S phase, and 27.7% in G2/M phase (Figure 5B). However, in the presence of isookanin, the majority of cells stayed in the S phase and thus did not enter the G2/M phase. These results suggest that the antiangiogenic effect of isookanin was associated with cell cycle control. Furthermore, the flow cytometric data did not detect the increase of sub-G1 cells after isookanin treatment compared with the control group; thus, the possibility of cell death induced by isookanin under this experimental condition was ruled out again. In addition, we performed Western blot analysis for cell cycle-related proteins (Cyclin A_2_, Cyclin B_1_, Cyclin D_1_). Figure 5C shows that treatment with isookanin significantly decreased the expression of Cyclin A_2_ and Cyclin D_1_ and that treatment slightly reduced the expression of Cyclin B_1_. These results indicate that isookanin inhibited cell proliferation by downregulating the expression of cell cycle-related proteins to arrest the S phase cell.

### 2.6. Effects of Isookanin on Phophorylation of ERK1/2 and CREB in PGE_2_-Induced HMEC-1 Cells

Mitogen-activated protein kinase (MAPK) pathways constitute a large modular network that regulates a variety of physiological processes, including cell growth and differentiation. Extracellular signal regulated kinases (ERK1/2) are responsible for stimulating numerous downstream effectors, many of which are transcription factors [33]. Cyclic adenosine monophosphate (cAMP) response element-binding protein (CREB), is one of the transcription factors that plays an important role in cell proliferation and the cell cycle [34]. To characterize whether isookanin acts on the PGE_2_-mediated angiogenesis signaling pathway, activation of ERK1/2 and CREB was investigated. Figure 6 shows that the phosphorylation of ERK1/2 and CREB was induced by PGE_2_ (20 µM), and phosphorylation of both ERK1/2 and CREB was inhibited by isookanin. These results indicate that isookanin inhibited ERK1/2 phosphorylation, which may further suppress CREB activation. Additionally, since CREB activation can be achieved by cAMP/PKA activation in an ERK-independent manner [35], the possibility that isookanin directly inhibits CREB phosphorylation via ERK-independent pathway cannot be excluded. In this regard, it is necessary to confirm through additional experiments whether isookanin inhibits CREB phosphorylation directly.

## 3. Discussion

Inflammatory mediators are well known for their role in inducing an appropriate immune response to external stimuli, and recently, they have also been shown to play an important role in the body’s response to vascular disease [36]. Previous studies have shown that the inflammatory response enhances angiogenesis and tumor growth [37]. Prostaglandins, especially PGE_2_, have been demonstrated to play an important role in immune regulation, and according to recent reports, PGE_2_ also plays a role in promoting the assembly of new blood vessels in angiogenesis [14]. The development of anti-angiogenesis agents has attracted great interest due to the high prevalence of angiogenesis-related diseases. The importance of antiangiogenic agents in the treatment of cancer is well established [38], and their role in the treatment of chronic inflammatory disorders is also gaining more support. In cases such as rheumatoid arthritis [39] and human IBD [40], it has been demonstrated that angiogenesis plays an important role in the pathophysiology of these chronic inflammatory diseases. There are several types of synthetic drugs for anti-angiogenesis therapy, but considering their side effects, it is worth investigating bioactive active compounds derived from natural products for the prevention and treatment of angiogenesis-related diseases [41]. Currently, a large variety of polyphenolic compounds from natural sources have drawn considerable attention for their multiplex biologic properties. Many natural compounds—for example, resveratrol, curcumin, or quercetin—can target a wide variety of molecules, implying that natural products might be effective inhibitors [42]. Isookanin is a phenolic flavonoid component contained in some plants, such as *Asteraceae* [26] and *Leguminosae* [28], and its anti-inflammatory efficacy and mechanism of action have already been revealed in our previous research [29].

In this study, we investigated the role that isookanin plays in the process of PGE_2_-induced angiogenesis. The results of the LDH experiment show that isookanin has no cytotoxicity, and the results of MTT cell viability confirm that isookanin can reduce the proliferation of PGE_2_-induced HMEC-1 cells in a concentration-dependent manner, while not causing cell damage. In addition, we confirmed that isookanin inhibited the migration and tube formation of HMEC-1 cells induced by PGE_2_. These results support the potential of isookanin as an antiangiogenic agent. In addition, isookanin induces the S phase arrest of HMEC-1 cells through downregulating the expression of cell cycle-related proteins. Moreover, the flow cytometric data of cell cycle assay did not detect the increase of sub-G1 cells after isookanin treatment compared with the control group. This indicates that the antiangiogenic activity of isookanin is due to the inhibition of cell proliferation, not the induction of cell death.

PGE_2_ affects target cells by activating cognate receptors EP1-4 that belong to the G protein-coupled receptor superfamily of cell surface-expressed receptors [15]. The receptors demonstrate distinct, as well as opposing, effects on intracellular signaling events. The EP1 receptor couples to Gq and mediates a rise in intracellular calcium concentration, and the EP3 receptor couples to Gi, reducing cAMP concentration. Both EP2 and EP4 couple to Gs, leading to increased synthesis of cAMP and consequent activation of PKA [14]. Subsequent PKA activation is involved in multiple signaling, including activation of the cAMP response element (CREB) [35]. The transcription factor CREB plays a key role in controlling cell growth, survival, and cell cycle progression, and it plays an important role in the development of many cell types, including vascular smooth muscle cells [43], adipocytes [44], glioma cell [45], and vascular endothelial cells [46]. Multiple intracellular signaling kinases can induce the phosphorylation of CREB, including extracellular signal regulated kinases (ERKs) [45,47]. Moreover, the ERK signaling cascade is a central MAPK pathway that plays a role in the regulation of various cellular processes, such as proliferation, differentiation, survival, and apoptosis under some conditions [48]. For example, Pozzi et al. [49] demonstrated an EP4-mediated PI3K/ERK signaling pathway in mouse colon carcinoma cells. Activation of the ERK1/2 pathway is required for the growth, proliferation, and migration of endothelial cells, and this pathway is considered an important target for antiangiogenic agents [50,51].

In recent studies, it is known that PGE_2_ can promote cell growth by activating multiple signaling pathways involved in cell proliferation and that both MAPK and cAMP/PKA pathways are required for PGE_2_-induced cell proliferation [45,52]. In this study, PGE_2_ was shown to increase phosphorylation of ERK1/2 and CREB; therefore, PGE_2_-induced proliferation is associated with activation of ERK1/2 and CREB in vascular endothelial cells. In addition, isookanin exhibits the efficacy of inhibiting phosphorylation of both ERK 1/2 and CREB induced by PGE_2_. These results suggest that the mechanism by which isookanin inhibits the proliferation of vascular endothelial cells is by inhibition of ERK1/2 and CREB phosphorylation. However, these results should be confirmed by further experiments to determine whether isookanin inhibits CREB activation through ERK1/2 dependence or acts directly on CREB.

PGE_2_ may induce angiogenesis by acting indirectly on a variety of cell types to produce proangiogenic factors. Indeed, PGE_2_ stimulation of airway smooth muscle, endometrium, or colon cancer cells leads to increased production of VEGF, bFGF, or CXCL1 that, in turn, act on target endothelial cells to promote the angiogenesis [53,54,55]. In addition, previous studies have reported that PGE_2_ acts directly on endothelial cells, promoting the creation of new blood vessels through activation of G protein-coupled receptor EP4 and its downstream pathway [14]. Of all the PGE_2_ receptors, the roles of the EP4 receptor in health and disease have received much attention [56]. Chell et al. [57] reported that EP4 receptor protein expression was increased in colorectal cancers, as well as in adenomas when compared with normal colonic epithelium. EP receptor overexpression—and thus upregulation of various signaling cascades associated with angiogenic tumorigenesis, PGE_2_ modification of the tumor microenvironment, and evasion of the immune system—is regulated through the EP receptors [58]. Therefore, the direct effect of isookanin on the expression of the EP receptor was confirmed and is presented in Appendix A. The result shows that isookanin effectively inhibited the expression of EP4 (Appendix A). These results suggest that isookanin possibly regulates downstream signaling by inhibiting the expression of EP4, and thereby inhibiting angiogenesis. However, to confirm the effect between the suppression of EP4 expression and downstream signaling, additional experiments are required. In addition, recent studies have reported that several specific receptors may be involved in the angiogenesis of PGE_2_. Finetti et al. [59] reported that PGE_2_ primes the angiogenic switch through a synergistic interaction with the fibroblast growth factor-2 pathway. There is also a report claiming that EP2 receptor agonists stimulate angiogenesis to promote adipogenesis [60]. Based on these studies, it can be expected that several specific receptors, including EP4, are the major receptors responsible for PGE_2_-induced angiogenesis. To identify the role of specific receptors for PGE_2_ in an angiogenesis model, further studies using specific EP antagonists or gene silencing are required.

## 4. Materials and Methods

### 4.1. Reagents

MCDB 131 medium, fetal bovine serum, L-glutamine, and epidermal growth factor (EGF) were purchased from Gibco (Grand Island, NY, USA). Prostaglandin E_2_ (PGE_2_), hydrocortisone, dimethyl sulfoxide (DMSO), and propidium iodide (PI) were purchased from Sigma-Aldrich (St. Louis, MO, USA), and human recombinant VEGF was obtained from ProSpec (Rehobot, Israel). Isookanin (purity ≥ 98%) was obtained from ChemFaces Biochemical Co. Ltd. (Wuhan, China) and dissolved in DMSO. Antibodies for phospho CREB and total CREB, phospho ERK1/2 and total ERK1/2, and cell cycle-related antibodies (Cyclin A_2_, Cyclin B_1_, Cyclin D_1_) were purchased from Cell Signaling Technology, Inc. (Danvers, MA, USA), while β-actin and secondary antibodies were purchased from Santa Cruz Biotechnology, Inc. (Dallas, TX, USA).

### 4.2. Cell Culture

The human dermal microvascular endothelial cell line (HMEC-1) was obtained from the American Type Culture Collection (ATCC, Manassas, VA, USA). Cells were cultured in MCDB 131 medium supplemented with 10% fetal bovine serum, 10 mM L-glutamine, 10 ng/mL epidermal growth factor (EGF), and 1 µg/mL hydrocortisone. Cells were maintained at 37 °C in a humidified atmosphere of 5% CO_2_.

### 4.3. Cell Proliferation Assay

The effect of isookanin on cell proliferation was determined via MTT (Duchefa, Haarlem, The Netherlands) assay. HMEC-1 cells were plated at a density of 4 × 10^4^ cells/well in 24-well plates and incubated overnight. Then, the cells were pre-treated 2 h in the presence or absence of isookanin (1, 5, and 10 µg/mL) in serum-free medium and stimulated by PGE_2_ (20 µM) for 48 h. VEGF (100 ng/mL) was used as a positive control to support that PGE_2_ is a potent proangiogenic factor, and an untreated group was used as a control. After treatment, microscopic observation was performed using an inverted microscope (Nikon Inc., Japan), and then, MTT reagent (1 mg/mL) was added to each well and incubated for 3 h before the supernatants were discarded followed by the addition of 200 μL DMSO. The absorbance at 570 nm was measured using a microplate reader (BioTek Instruments, Inc., Winooski, VT, USA). The results are expressed as a percentage of control.

### 4.4. Lactate Dehydrogenase (LDH) Cytotoxicity Assay

The LDH release assay was performed using a LDH cytotoxicity detection kit (Enzo Life Sciences, Inc., Farmingdale, NY, USA) according to the manufacturer’s instructions. In brief, HMEC-1 cells were seeded in 24-well plates at a density of 5 × 10^4^ cells per well. The cells were then incubated with various concentrations of isookanin (1, 5, and 10 µg/mL) for 48 h, or they were incubated with lysis buffer as a high cytotoxicity control. The cell supernatants were collected and analyzed. The absorbance of formed formazan was read at 490 nm on a microplate reader (BioTek Instruments, Inc., Winooski, VT, USA), and the results are expressed as a percentage of control.

### 4.5. Scratch Migration Assay

To assess the effect of isookanin on HMEC-1 growth following angiogenic stimuli, a migration assay was performed as described previously [61]. Briefly, HMEC-1 cells were seeded in 12-well plates at a density of 2 × 10^5^ cells/well in 10% serum containing MCDB 131 medium for 24 h. After the cells reached 90% confluence, the cells were scratched with a sterile pipette tip and washed with phosphate buffered saline (PBS) subsequently to eliminate the impaired cells. The cells were then incubated in 1% fetal bovine serum medium containing different concentrations of isookanin (1, 5, and 10 µg/mL) and stimulated with PGE_2_ (20 µM) or VEGF (100 ng/mL) for 24 h, and an untreated group was used as a control. The migration of HMEC-1 cells across the demarcation line was monitored using an inverted microscope (Nikon Inc., Japan).

### 4.6. Transwell Migration Assay

A transwell analysis was performed to observe cell migration in 3D models through a transwell Boyden Chamber (8.0 μM pore size, Corning Costar Corporation, NY, USA), which were coated with 1% gelatin. The HMEC-1 cell suspensions were seeded at a density of 1 × 10^5^ cells/well into the upper chambers of the Boyden system and cultured in 1% FBS medium containing different concentrations of isookanin (1, 5, 10 μg/mL) or the solvent control [61,62]. The lower chamber added 700 μL serum-free medium containing 20 µM PGE_2_ or 100 ng/mL VEGF to induce angiogenesis, and the untreated group was used as a control. Twenty-four hours later, all non-migrant cells in the upper chamber were removed using a cotton swab. The chambers were washed twice with Dulbecco’s phosphate-buffered saline (DPBS) and fixed with 4% paraformaldehyde for 30 min. After washing twice with DPBS again, the chambers were stained with 5% crystal violet stain for 20 min, and then the stained cells were photographed under a microscope. Cells were subsequently extracted with 10% acetic acid; the absorbance at 600 nm was used to calculate the percentage of migrated cells.

### 4.7. Matrigel Endothelial Cell Tube Formation Assay

Tube formation assay was performed, as previously described, to analyze in vitro angiogenesis [61]. Briefly, a pre-cooled 24-well plate was coated with Matrigel (Corning Costar Corporation, NY, USA) and allowed to solidify at 37 °C for 1 h. After coagulation, HMEC-1 cells (1.5 × 10^4^ cells/well) were seeded on the Matrigel and cultured in 1% FBS medium containing different concentrations of isookanin (1, 5, and 10 μg/mL), and 20 µM PGE_2_ or 100 ng/mL VEGF was added to stimulate endothelial cell tube formation, while control wells were not treated with any test agent. The angiogenesis assay plates were then incubated for 24 h at 37 °C in a 5% CO_2_ atmosphere.

Following incubation and removal of medium from the plates, Calcein AM dye diluted to 8 μg/mL concentration in Hanks Balanced Salt Solution (HBSS) was added to each well, and the plates were incubated at 37 °C with 5% CO_2_ for 40 min to label the cells. After incubation, the plates were photographed using an EVOS fluorescent microscope (Advanced Microscopy Group, Bothell, WA, USA). Tube formation was evaluated by analyzing the total branch number, length, and mesh size of the tube-like structures using Image-J software (National Institutes of Health, Bethesda, MD, USA).

### 4.8. Flow Cytometry Analysis of Edu Cell Proliferation Assay and Cell Cycle Arrest

Cell proliferation was measured using the Click-iT EdU Cytometry Assay Kit (Invitrogen, Carlsbad, CA, USA) and propidium iodide dual staining according to the manufacturer’s instructions. HMEC-1 cells were seeded into 6-well plates at 1.2 × 10^5^ per well and incubated overnight. The cells were pretreated with or without isookanin (1, 5, and 10 μg/mL) for 2 h and stimulated with 20 µM PGE_2_ or 100 ng/mL VEGF in each well for 24 h under serum-free conditions. The cells were then labeled with EdU (10 μM) for 3 h, fixed for 15 min with 4% paraformaldehyde, and permeabilized for 5 min at room temperature using a Triton X-100 solution. The fixed cells were then stained with the Click-iT reaction mixture for 30 min at room temperature. Afterwards, propidium iodide-staining was performed to determine cell proliferation and cell cycle distribution, and the cells were analyzed with BD FACS Calibur™ flow cytometry and BD CellQuest™ Pro Software (Becton Dickinson, San Jose, CA, USA).

### 4.9. Western Blotting

Western blotting was performed to measure the levels of proteins associated with the cell cycle and angiogenic signaling pathways. HMEC-1 cells were pretreated with or without isookanin (1, 5, and 10 μg/mL) for 2 h prior to induction with 20 µM PGE_2_. The cells were then harvested and lysed by protein extraction solution, PRO-PREP™ (iNtRON Biotechnology, Seongnam, Korea). The protein extracts (20 μg) from lysed cells were loaded onto a NuPAGE Novex 10% Bis-Tris Gel 1.0 mm, 15-well (Invitrogen, Waltham, MA, USA), and transferred to a nitrocellulose membrane by using an iBlot™ Gel Transfer Device (Invitrogen, Waltham, MA, USA). The membranes were blocked with 5% bovine serum albumin (BSA) for 1 h. Membranes were then incubated with primary antibodies (1:1000 dilution of stock) in 10 mL of blocking buffer, followed by incubation with horseradish peroxidase-conjugated secondary antibody (1:3000 dilution of stock). The protein bands were detected by enhanced chemiluminescence detection reagents (Invitrogen, Waltham, MA, USA) and visualized using ImageQuant™ LAS 500 (Cytiva, Marlborough, MA, USA).

### 4.10. Statistical Analyses

All experimental data are expressed as mean ± standard deviation. Differences between the control and the treatment group were evaluated by one-way ANOVA (SPSS, IBM, Armonk, NY, USA). Values of *p* < 0.05, 0.01, or 0.001 were considered statistically significant.

## 5. Conclusions

In this study, we demonstrated that isookanin has an effective inhibitory effect on the angiogenic properties of microvascular endothelial cells. Isookanin exerts inhibitory effects at multiple stages of PGE_2_-induced angiogenesis, including the growth, proliferation, migration, and tube formation of microvascular endothelial cells. In addition, isookanin induces cell cycle arrest in S phase, which is also the reason for subsequent inhibition of cell proliferation. Our results also suggest that the mechanism of inhibiting angiogenesis by isookanin is derived from the inhibition of PGE_2_-mediated phosphorylation of ERK1/2 and CREB (Figure 7). Based on these results, isookanin shows its potential as a candidate for the treatment of angiogenesis-related diseases, and it is necessary to further determine the therapeutic effect of isookanin as an antiangiogenic agent in established animal models and clinical trials.

## Figures and Tables

**Figure 1 ijms-22-06466-f001:**
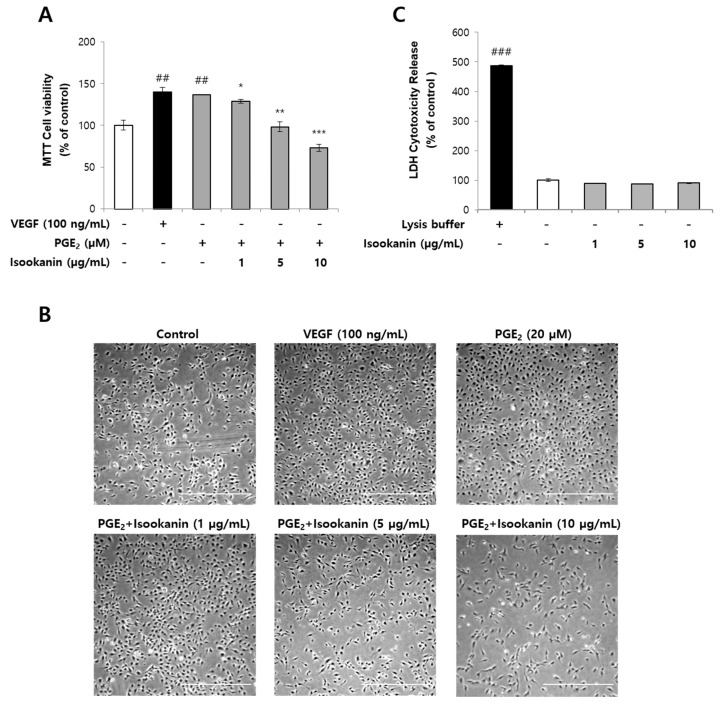
The effect of isookanin on cell proliferation in PGE_2_-induced HMEC-1 cells. Cells were pretreated with the indicated concentrations of isookanin for 2 h before stimulation with PGE_2_ (20 µM) or VEGF (100 ng/mL) for 48 h. (**A**) Cell viability was measured using the MTT assay. (**B**) The density of endothelial cells was observed under a microscope; scale bars are 80 μm. (**C**) Cytotoxicity was measured using the lactate dehydrogenase (LDH) cytotoxicity assay. The results are mean ± standard deviation (SD) (*n* = 3). ^##^ *p* < 0.01, ^###^ *p* < 0.001 vs. control. * *p* < 0.05, ** *p* < 0.01, and *** *p* < 0.001 vs. PGE_2_-treated control.

**Figure 2 ijms-22-06466-f002:**
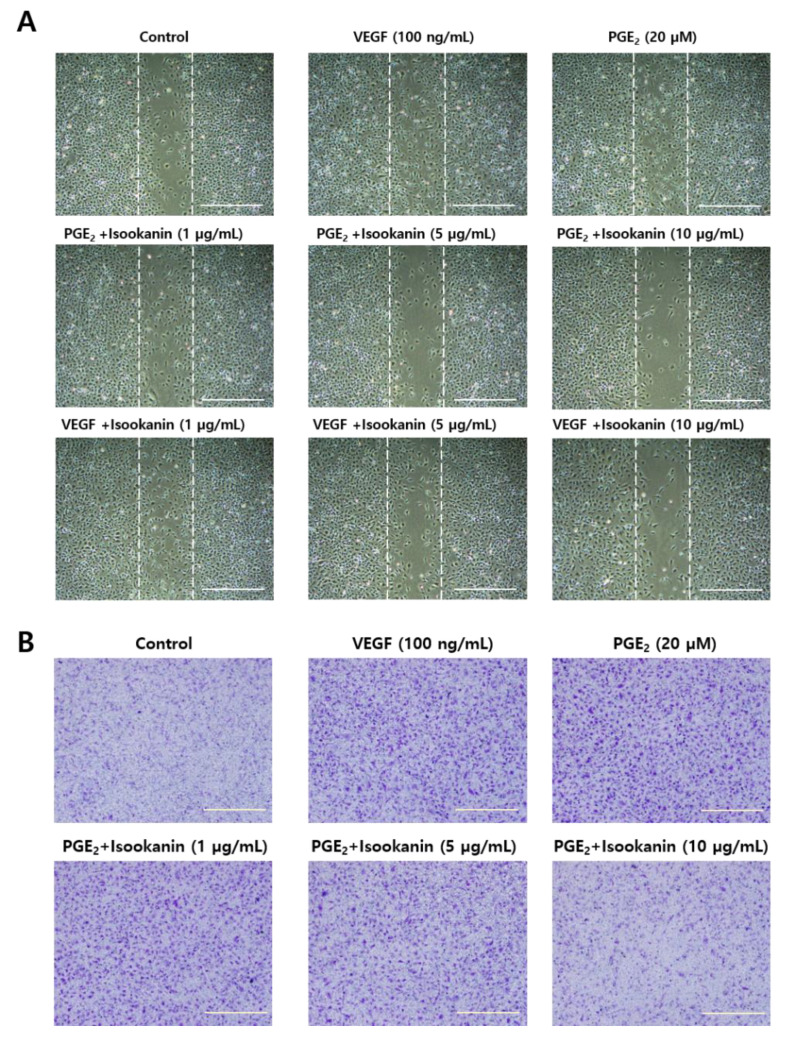
The effect of isookanin on cell migration in PGE_2_-induced HMEC-1 cells. Cells were pretreated with the indicated concentrations of isookanin for 2 h before stimulation with PGE_2_ (20 µM) or VEGF (100 ng/mL) for 24 h. (**A**) Cell migration was measured using the scratch migration assay, and cells were photographed under a microscope; scale bars are 80 µm. (**B**) Cell migration was measured using the transwell migration assay, and cells were photographed under a microscope; scale bars are 200 µm. (**C**) Cells were extracted with 10% acetic acid, the absorbance at 600 nm was used to calculate the percentage of migrated cells. The results are mean ± standard deviation (SD) (*n* = 3). ^#^ *p* < 0.05 vs. control. * *p* < 0.05 vs. PGE_2_-treated control.

**Figure 3 ijms-22-06466-f003:**
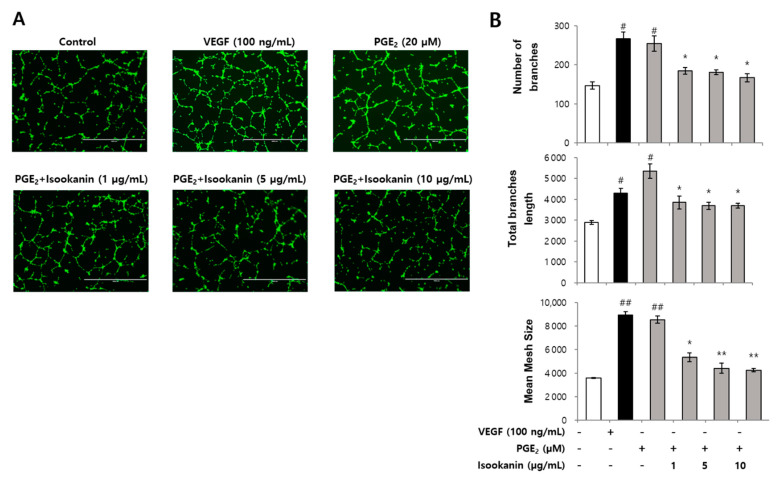
The effect of isookanin on tube formation in PGE_2_-induced HMEC-1 cells. Cells were pretreated with the indicated concentrations of isookanin for 2 h before stimulation with PGE_2_ (20 µM) or VEGF (100 ng/mL) for 24 h. HMEC-1 cells tube formation was measured using the tube formation assay. (**A**) Calcein AM dye-stained cells were photographed under a microscope; scale bars are 1000 μm. (**B**) Tube formation was evaluated by analyzing the total branch number, length, and mesh size of the tube-like structures using Image-J software. The results are mean ± standard deviation (SD) (*n* = 3). ^#^ *p* < 0.05, ^##^ *p* < 0.01 vs. control. * *p* < 0.05, ** *p* < 0.01 vs. PGE_2_-treated control.

**Figure 4 ijms-22-06466-f004:**
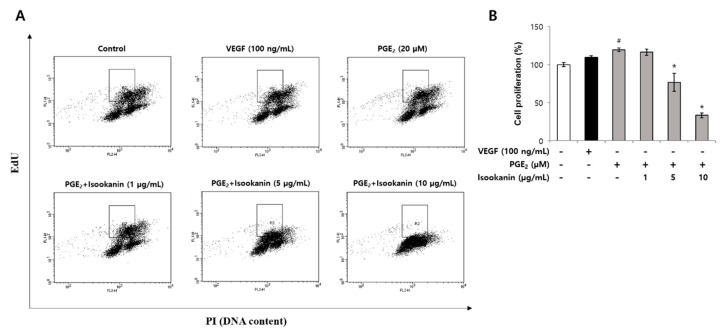
The anti-proliferative effect of isookanin on PGE_2_-induced HMEC-1 cells. Cells were pretreated with the indicated concentrations of isookanin for 2 h before stimulation with PGE_2_ (20 µM) for 24 h. Then, cell proliferation was measured using the Click-iT EdU Cytometry Assay Kit and propidium iodide dual staining to determine cell proliferation. (**A**) EdU and DNA contents were analyzed by flow cytometry. Bivariate distributions of EdU incorporation (*y* axis) vs. DNA content (*x* axis) were plotted. The boxes indicate EdU-incorporated cells in the S-phase. (**B**) Cell proliferations are expressed as percentages of EdU-labeled S-phase populations to total cell numbers. The results are mean ± standard deviation (SD) (*n* = 3). ^#^ *p* < 0.05 vs. control. * *p* < 0.05 vs. PGE_2_-treated control.

**Figure 5 ijms-22-06466-f005:**
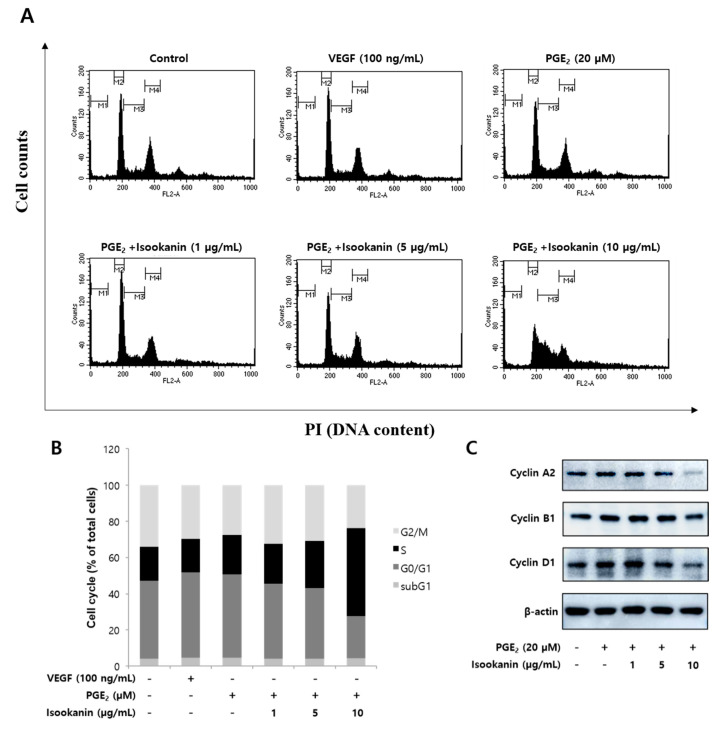
Effects of isookanin on the regulation of cell cycle and the expression level of cell cycle regulatory proteins on PGE_2_-induced HMEC-1 cells. Cells were pretreated with the indicated concentrations of isookanin for 2 h before stimulation with PGE_2_ (20 µM). (**A**) Total DNAs were labeled with propidium iodide and analyzed by flow cytometry. (**B**) The cell cycle distribution was analyzed with BD CellQuest™ Pro Software. (**C**) Cell cycle-related proteins (Cyclin A_2_, Cyclin B_1_, Cyclin D_1_) and β-actin (a loading control) were analyzed by Western blotting.

**Figure 6 ijms-22-06466-f006:**
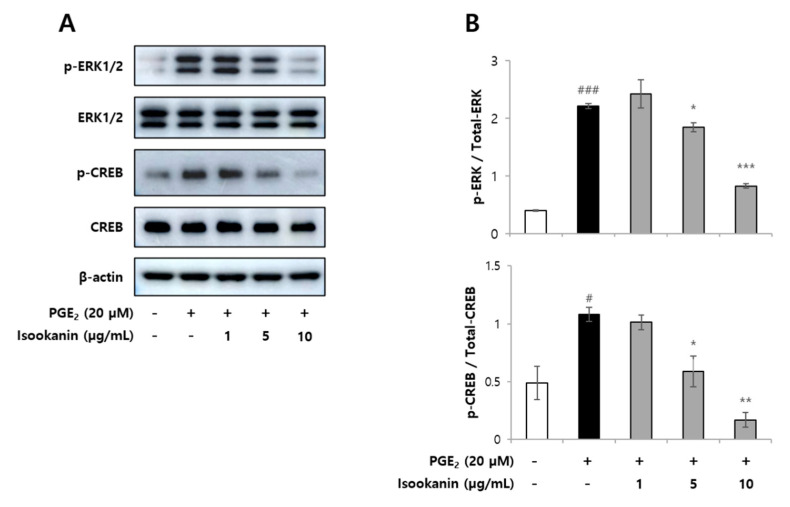
Effects of isookanin on the phosphorylation of ERK and CREB in PGE_2_-induced HMEC-1 cells. The HMEC-1 cells were cultured for 24 h and then treated with PGE_2_ (20 µM) in the presence or absence of isookanin. Cell lysates were prepared at 20 min and were then subjected to Western blot analysis. (**A**) The bands for phospho-ERK1/2 and phospho-CREB were detected and normalized to their total forms of ERK1/2 and CREB. (**B**) The scanning densitometric values of each band were analyzed with Image J. The results are mean ± standard deviation (SD) (*n* = 3). ^#^ *p* < 0.05, ^###^ *p* < 0.001 vs. control. * *p* < 0.05, ** *p* < 0.01, *** *p* < 0.001 vs. PGE_2_-treated control.

**Figure 7 ijms-22-06466-f007:**
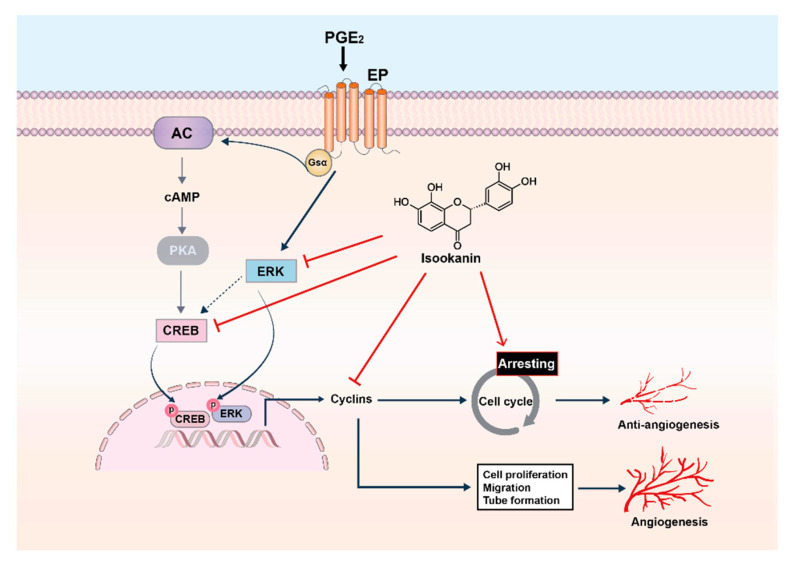
The mechanism of isookanin inhibiting angiogenesis is associated with suppressing PGE_2_-mediated ERK1/2 and CREB phosphorylation and with inducing cell cycle arrest.

## Data Availability

The data presented in this study are available on request from the corresponding author.

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
