# Peer review of "Isookanin Inhibits PGE2-Mediated Angiogenesis by Inducing Cell Arrest through Inhibiting the Phosphorylation of ERK1/2 and CREB in HMEC-1 Cells"

_ijms, 2021, doi:10.3390/ijms22126466_

Round 1
Reviewer 1 Report
Practical investigation is reliable. My only concern is the use of a single cell line and not knowing how representative is that of all endothelial cells.Author Response
Practical investigation is reliable. My only concern is the use of a single cell line and not knowing how representative is that of all endothelial cells.
- Thanks for your comment. In this experiment, only human microvascular endothelial cells (HMEC-1) were used to confirm the effect of isookanine on angiogenesis by PGE2. Since HMEC-1 cells are mainly located in capillaries, this cell line was used because it was expected to be most suitable for skin erythema model study. In further study, we plan to check the effect of isookanin by applying it to various vascular endothelial cells.
Reviewer 2 Report
The manuscript of Xin and colleagues investigated the role of a natural product, isookanin, on the angiogenic activity of PGE2, a prostanoid involved in vascular inflammation. The topic is interesting and the control of angiogenesis in inflammatory diseases is a medical need. However, in the present form the manuscript needs a substantial revision.
- The concentration of PGE2 used is very high, even to mimic an inflammatory environment. Why did the authors choose such a high dose?
- Figure 1 and 2: Please, report a bar scale or magnification in figure legend.
- Figure 3: Please, report the measure of bar scale in figure legend.
- Figure 6: PGE2 binds EP4 and induces an increase in cAMP and activation of PKA. Why do the authors measure EP4 expression? Receptor expression is not indicative of activation of PGE2/EP4 complex signaling.
- Figure 6 is qualitative but of low quality. To claim that EP4 mediates the angiogenic activity of PGE2 and that isookanin inhibits the angiogenic activity of PGE2 by reducing EP4 expression levels, it is necessary for the authors to demonstrate that EP4 is responsible for the activity of PGE2 (does gene silencing or pharmacological inhibition of EP4 inhibit the proangiogenic activity of PGE2 on the cell model used)?
- expression of pCREB is convincing, that of pERK1/2 is of low quality. You need to replace it with a better image. Are these blots representative of how many experiments? Can the data be quantified, and statistical analysis done? Statistical analysis would strengthen the result.
- The signaling pathway in Figure 8 is not supported by the results: the authors do not demonstrate that PKA is responsible for ERK activation. CREB activation could be dependent on PKA and ERK via two independent pathways, both activated by EP4.
- The authors do not demonstrate that EP4 is the receptor responsible for the signaling pathway activated by PGE2. The experiments are not convincing because they lack controls: a) do HMEC-1 cells express only EP4, or do they express all EP receptors? b) Does Isookanin inhibit the expression of EP4 only? What effect does it have on other receptors?
- Transwell migration assay: migration was measured after 12 h. What are the expression levels of EP4 in HMEC-1 after 12 h of stimulation with PGE2? At the molecular level, how does PGE2 stimulate expression of its receptor? Through what signaling? Similar for tube formation assay (18 h of PGE2 stimulation).
- There are interesting manuscripts in which the receptor responsible for the angiogenic properties of PGE2 is not EP4 (Prostaglandins Other Lipid Mediat. 2014; 112:9-15. doi: 10.1016/j.prostaglandins.2014.05.003. PMID: 24911647; Circ Res. 2009 Sep 25;105(7):657-66. doi: 10.1161/CIRCRESAHA.109.203760. PMID: 19713533; Angiogenesis. 2005;8(4):339-48. doi: 10.1007/s10456-005-9023-8. PMID: 16400521). Please, quote complete literature in the manuscript and discuss this aspect.
Author Response
The manuscript of Xin and colleagues investigated the role of a natural product, isookanin, on the angiogenic activity of PGE2, a prostanoid involved in vascular inflammation. The topic is interesting and the control of angiogenesis in inflammatory diseases is a medical need. However, in the present form the manuscript needs a substantial revision.
We greatly appreciate for your constructive suggestions. They have helped us improve the quality of the manuscript. The comments are listed below with our responses.
- The concentration of PGE2 used is very high, even to mimic an inflammatory environment. Why did the authors choose such a high dose?
- As you pointed out, the concentration of PGE2 used in in vitro experiments is too high to mimic the inflammatory environment. However, this experimental model did not show cellular reactivity at low concentrations of PGE2. PGE2 is a well-known inflammation inducer. Therefore, to construct a PGE2-induced inflammatory erythema model, it was necessary to determine the optimal treatment concentration. In this experimental model, since the optimal cell reactivity was exhibited at a concentration of 20 µM, the experiment was performed with a high concentration of PGE2. Although the concentration of PGE2 treated in this experiment was high, it was possible to study the induction of angiogenesis and its mechanism by PGE2. Therefore, our experimental model is considered sufficient to mimic the inflammatory erythema environment.
- Figure 1 and 2: Please, report a bar scale or magnification in figure legend.
- A scale bar was added to figure 1 and figure 2, and the magnification was described in figure legend.
(Figure 1: page 3, line 109, 113) and (Figure 2: page 4, line 130, page 4, line 134-136)
- A scale bar was added to figure 1 and figure 2, and the magnification was described in figure legend.
- Figure 3: Please, report the measure of bar scale in figure legend.
- A description of the magnification has been added to the figure legend.
(Page 6, line 157)
- Figure 6: PGE2 binds EP4 and induces an increase in cAMP and activation of PKA. Why do the authors measure EP4 expression? Receptor expression is not indicative of activation of PGE2/EP4 complex signaling.
- EP receptor overexpression and thus upregulation of various signaling cascades associated with angiogenic tumorigenesis, PGE2 modification of the tumor microenvironment and evasion of the immune system is regulated through the EP receptors. Therefore, before confirming the effect of isookanin on angiogenic signaling factors (cAMP, PKA, etc.) mediated by the binding of the PGE2/EP receptor, the direct effect on the expression of the EP receptor was first confirmed. As a result, it showed the result that isookanin effectively inhibited the expression of EP4. These results are possible to predict that isookanin can regulate downstream signaling by inhibiting the expression of EP4 and thereby inhibit angiogenesis. However, to confirm the effect between the suppression of EP4 expression and downstream signaling, additional experiments are required. Since the direct relationship between EP receptor expression and angiogenesis needs to be confirmed through further experiments, these results have been moved to the Supplementary Information section.
(Page 11, line 302-314)
- Figure 6 is qualitative but of low quality. To claim that EP4 mediates the angiogenic activity of PGE2 and that isookanin inhibits the angiogenic activity of PGE2 by reducing EP4 expression levels, it is necessary for the authors to demonstrate that EP4 is responsible for the activity of PGE2 (does gene silencing or pharmacological inhibition of EP4 inhibit the proangiogenic activity of PGE2 on the cell model used)?
- In this study, only the EP4 expression inhibitory effect of isookanin was confirmed, so additional experiments are needed to confirm the association between EP4 expression and angiogenesis. As you suggested, we plan to confirm the specific responsibility of EP4 in PGE2-derived angiogenesis through EP4 antagonist or gene silencing experiments. Therefore, Figure 6 has been included to supplement the experiment, and moved to the Supplementary Information section.
(Page 11, line 302-314)
- In this study, only the EP4 expression inhibitory effect of isookanin was confirmed, so additional experiments are needed to confirm the association between EP4 expression and angiogenesis. As you suggested, we plan to confirm the specific responsibility of EP4 in PGE2-derived angiogenesis through EP4 antagonist or gene silencing experiments. Therefore, Figure 6 has been included to supplement the experiment, and moved to the Supplementary Information section.
- expression of pCREB is convincing, that of pERK1/2 is of low quality. You need to replace it with a better image. Are these blots representative of how many experiments? Can the data be quantified, and statistical analysis done? Statistical analysis would strengthen the result.
- phospho-ERK 1/2 blot image were replaced with good quality image. Statistical analysis was performed for each blot through an image analysis program (ImageJ). The significance of the experiment was confirmed by statistical analysis, and the results are presented in a graph (Figure 6B).
(Page 9, line 222-229)
- The signaling pathway in Figure 8 is not supported by the results: the authors do not demonstrate that PKA is responsible for ERK activation. CREB activation could be dependent on PKA and ERK via two independent pathways, both activated by EP4.
- CREB activation may depend on PKA and ERK through two independent pathways activated by EP4. Our experimental results showed that isookanin inhibited p-ERK1/2 and p-CREB, but we did not confirm whether the phosphorylation of CREB was regulated by ERK-dependent or independent manner. Therefore, Figure 7 has been modified and will be confirmed through further experiments.
(Page 9, line 215-221 and Page 11, line 285-3295) and (Page 12, line 324)
- The authors do not demonstrate that EP4 is the receptor responsible for the signaling pathway activated by PGE2. The experiments are not convincing because they lack controls: a) do HMEC-1 cells express only EP4, or do they express all EP receptors? b) Does Isookanin inhibit the expression of EP4 only? What effect does it have on other receptors?
- As you suggested, we did not demonstrate that EP4 is the only receptor responsible for the signaling pathway of angiogenesis activated by PGE2. In addition, the effect of isookanin on other EP receptors was not confirmed. In this study, we suggested that EP4 is a major receptor for PGE2-induced angiogenesis, and additional experiments are required to confirm the receptor specificity of PGE2 in angiogenic signaling pathway.
(Page 11, line 308-323)
- As you suggested, we did not demonstrate that EP4 is the only receptor responsible for the signaling pathway of angiogenesis activated by PGE2. In addition, the effect of isookanin on other EP receptors was not confirmed. In this study, we suggested that EP4 is a major receptor for PGE2-induced angiogenesis, and additional experiments are required to confirm the receptor specificity of PGE2 in angiogenic signaling pathway.
- Transwell migration assay: migration was measured after 12 h. What are the expression levels of EP4 in HMEC-1 after 12 h of stimulation with PGE2? At the molecular level, how does PGE2 stimulate expression of its receptor? Through what signaling? Similar for tube formation assay (18 h of PGE2 stimulation).
- This part was written in error while writing the experimental method. The figure legends for each experiment are all written correctly. Transwell migration, scratch migration, and tube formation experiments are all results obtained after culturing for 24 hours after PGE2 EP4 expression only confirmed the relative expression at the protein level, not the expression at the gene level. Therefore, it was not quantified and expressed numerically. To confirm the change in the expression level of EP4 by the PGE2 treatment, it is necessary to check according to each incubation time.
(Page 13, line 371 and Page 14 line 386, Page 14 line 401)
- This part was written in error while writing the experimental method. The figure legends for each experiment are all written correctly. Transwell migration, scratch migration, and tube formation experiments are all results obtained after culturing for 24 hours after PGE2 EP4 expression only confirmed the relative expression at the protein level, not the expression at the gene level. Therefore, it was not quantified and expressed numerically. To confirm the change in the expression level of EP4 by the PGE2 treatment, it is necessary to check according to each incubation time.
- There are interesting manuscripts in which the receptor responsible for the angiogenic properties of PGE2 is not EP4 (Prostaglandins Other Lipid Mediat. 2014; 112:9-15. doi: 10.1016/j.prostaglandins.2014.05.003. PMID: 24911647; Circ Res. 2009 Sep 25;105(7):657-66. doi: 10.1161/CIRCRESAHA.109.203760. PMID: 19713533; Angiogenesis. 2005;8(4):339-48. doi: 10.1007/s10456-005-9023-8. PMID: 16400521). Please, quote complete literature in the manuscript and discuss this aspect.
- As you suggested, it is possible that several receptors are responsible for PGE2-induced angiogenesis. Regarding this, we mentioned the possibility that angiogenesis signaling by PGE2 can mediate receptors other than EP4 and added the content to the Discussion section. (Page 11, line 315-323)
Reviewer 3 Report
The title of the article" Isookanin inhibits PGE2-mediated angiogenesis by inducing cell arrest through the ERK/CREB signaling pathway in human dermal microvascular endothelial cells (HMEC-1) " corresponds to its content.
In the "Abstract" section, the authors briefly outlined the relevance of the search for new antiangiogenic drugs based on the phenolic flavonoid isoocanine, and presented their results and conclusion.
"Keywords" are optimal and reflect the main research direction.
In the chapter "Introduction", the authors fully and sufficiently described the problem and the importance of finding alternative strategies for controlling angiogenesis. The objective of the work is presented as a study of the effect of isoocanin on PGE2-induced angiogenesis in human skin microvessel endothelial cells (HMEC-1).
In the "Results" section, the authors presented the research results in full and in detail, and also substantiated the need to use one or another research method. The figures fully reflect the results obtained and are illustrative. All methods are presented in the Materials and Methods section. All necessary comments and links to other publications are provided. The article would look more attractive if the authors presented the study design. In the Discussion chapter, the authors discussed their results in a detailed and interesting way using the literature. The conclusions are convincing and consistent with the results.
Author Response
The title of the article" Isookanin inhibits PGE2-mediated angiogenesis by inducing cell arrest through the ERK/CREB signaling pathway in human dermal microvascular endothelial cells (HMEC-1) " corresponds to its content.
In the "Abstract" section, the authors briefly outlined the relevance of the search for new antiangiogenic drugs based on the phenolic flavonoid isookanine, and presented their results and conclusion.
"Keywords" are optimal and reflect the main research direction.
In the chapter "Introduction", the authors fully and sufficiently described the problem and the importance of finding alternative strategies for controlling angiogenesis. The objective of the work is presented as a study of the effect of isookanin on PGE2-induced angiogenesis in human skin microvessel endothelial cells (HMEC-1).
- Thanks for reviewing the paper.
We will further confirm the potential of isookanin as a treatment for inflammatory erythema through additional clinical efficacy trials.
Round 2
Reviewer 2 Report
Thank to the authors for the comprehensive responses and review of the manuscript. I have no further comments